SOFTWARE

# A software ecosystem for brain tractometry processing, analysis, and insight

John Kruper[1,2*], Adam Richie-Halford[3,4], Joanna Qiao[1,2], Asa Gilmore[1,2], Kelly Chang[1,2], Mareike Grotheer[5,6], Ethan Roy[3], Sendy Caffarra[7], Teresa Gomez[1,2], Sam Chou[1,2], Matthew Cieslak[8,9,10], Serge Koudoro[11], Eleftherios Garyfallidis[11], Theodore D. Satterthwaite[8,9,10], Jason D. Yeatman[3,4], Ariel Rokem[1,2*]

**1** Department of Psychology, University of Washington, Seattle, Washington, United States of America, **2** eScience Institute, University of Washington, Seattle, Washington, United States of America, **3** Graduate School of Education, Stanford University, Stanford, California, United States of America, **4** Division of Developmental-Behavioral Pediatrics, Stanford University, Stanford, California, United States of America, **5** Department of Psychology, Phillips-Universität Marburg, Marburg, Germany, **6** Center for Mind, Brain and Behavior, Phillips-Universität Marburg and Justus-Liebig Universität Giessen, Marburg, Germany, **7** Department of Biomedical, Metabolic and Neural Sciences, University of Modena and Reggio Emilia, Modena, Italy, **8** Penn/CHOP Lifespan Brain Institute, University of Pennsylvania, Philadelphia, Pennsylvania, United States of America, **9** Department of Psychiatry, University of Pennsylvania, Philadelphia, Pennsylvania, United States of America, **10** Penn Lifespan Informatics and Neuroimaging Center, University of Pennsylvania, Philadelphia, Pennsylvania, United States of America, **11** Luddy School of Informatics, Computing, and Engineering, Indiana University, Bloomington, Indiana, United States of America

* jk232@uw.edu (JK); arokem@uw.edu (AR)

**Data availability statement:** The software described in this paper is all fully available as open-source in https://tractometry.org. Datasets used herein are all publicly available.

## Abstract

Tractometry uses diffusion-weighted magnetic resonance imaging (dMRI) to assess physical properties of brain connections. Here, we present an integrative ecosystem of software that performs all steps of tractometry: post-processing of dMRI data, delineation of major white matter pathways, and modeling of the tissue properties within them. This ecosystem also provides a set of interoperable and extensible tools for visualization and interpretation of the results that extract insights from these measurements. These include novel machine learning and statistical analysis methods adapted to the characteristic structure of tract-based data. We benchmark the performance of these statistical analysis methods in different datasets and analysis tasks, including hypothesis testing on group differences and predictive analysis of subject age. We also demonstrate that computational advances implemented in the software offer orders of magnitude of acceleration. Taken together, these open-source software tools—freely available at https://tractometry.org—provide a transformative environment for the analysis of dMRI data.

## Author summary

The human brain is a highly inter-connected system. Information about the environment and about internal states is effectively distributed and integrated through neural

The Healthy Brain Network dataset and its derivatives are all available through the Amazon Web Services Open Data program's https://registry.opendata.aws/fcp-indi/. The Stanford HARDI dataset is available to donwload through the Stanford Data Repository: https://purl.stanford.edu/yx282xq2090. The infant dataset can be downloaded at: https://figshare.com/ndownloader/files/38053692. Data from Sarica et al. 2017 can be downloaded at https://yeatmanlab.github.io/Sarica_2017. A set of jupyter notebooks, which serve as computational companions to this article can be executed in full through Code Ocean at: https://doi.org/10.24433/CO.1278808.v2 and through Neurolibre at: https://preprint.neurolibre.org/10.55458/neurolibre.00037/.

**Funding:** This work was funded by National Institutes of Health grants RF1MH121868 (AR, JDY), RF1MH121867 (AR, TDS), R01EB027585 (AR, EG), R01MH113550 (TDS), R01MH112847 (TDS), R37MH125829 (TDS), R01MH120482 (TDS), and R01EB022573 (TDS), as well as by National Science Foundation grant 1934292 (AR). JK was supported through the NSF Graduate Research Fellowship DGE-2140004. Software development was also supported by the Chan Zuckerberg Initiative's Essential Open Source Software for Science program (SK, AR, EG), the Alfred P. Sloan Foundation and the Gordon & Betty Moore Foundation (AR). TDS and MC were additionally supported by the Penn-CHOP Lifespan Brain Institute, and the AE Foundation. MG was supported by the Deutsche Forschungsgemeinschaft (DFG, German Research Foundation)—project number 222641018—SFB/TRR 135 TP C10 as well as by "The Adaptive Mind", funded by the Excellence Program of the Hessian Ministry of Higher Education, Science, Research and Art. The funders had no role in study design, data collection and analysis, decision to publish, or preparation of the manuscript.

**Competing interests:** The authors have declared that no competing interests exist.

pathways that rapidly transmit signals between distant brain regions through large nerve fiber bundles. Measurements of diffusion MRI (dMRI) are sensitive to the trajectory of these nerve fiber pathways within the brain, and to their physical properties. We developed a suite of scalable open-source software tools that process dMRI data and delineate brain pathways and connections within it, quantifying the physical properties of brain tissue along the length of each pathway in an individualized manner. We demonstrate that the software is extensible to a variety of new studies, and offers useful approaches for visualization and statistical analysis, including novel machine learning tools. We also demonstrate that novel computational methods that we developed offer substantial speed-up, offering scalability for new large-scale datasets.

## Introduction

Diffusion-weighted Magnetic Resonance Imaging (dMRI) is the primary method currently used to conduct non-invasive measurements of living human brain connections. Because the diffusion of water within the tissue is restricted by tissue compartments, such as myelinated axons, this imaging technique provides detailed information about the trajectory of brain connections, as well as about their physical properties (e.g., density and organization) [1,2]. Therefore, dMRI is widely used in clinical and cognitive neuroscience studies, including in many large-scale consortium datasets (the Human Connectome Project, [3], the Adolescent Brain Cognitive Development study, [4], etc.). One of the challenges of analyzing dMRI data is that the details of the 3D structure of white matter anatomy vary between individuals. *Tractometry* is a methodology that addresses this challenge by identifying white matter pathways (tracts) in each individual's brain, and using these known anatomical structures as the coordinate system to compare measurements across subjects [5,6].

One approach to tractometry is Automated Fiber Quantification (AFQ) [7], which can be broken down into three steps. First, computational tractography methods are used to generate candidate pathways through the brain white matter. These candidate pathways contain both false positive and false negative connections [8], and suffer from the known biases of computational tractography [9]. Therefore, in the second step, candidate pathways are selected based on anatomical constraints that correspond to known tracts. This approach overcomes many of the concerns about confounds in dMRI-based tractography, because as previously shown, "brain connections derived from diffusion MRI tractography can be highly anatomically accurate – if we know where white matter pathways start, where they end, and where they do not go" [10]. Finally, microstructural brain tissue properties are inferred using a variety of different models [11–13] and their values along the length of each tract are extracted and mapped to one-dimensional "tract profiles". We have previously demonstrated that the AFQ approach to tractometry is reliable and robust [14].

Here, we present a comprehensive software ecosystem that supports end-to-end tractometry. The tools that make up this ecosystem emphasize interoperability and flexibility, allowing new, specialized analyses to be integrated with ease. Working within one framework reduces friction in data exchange and analysis, eases troubleshooting, and reduces the learning curve for new users. In developing these tools, we seek to use existing software and to rely on existing standards and best practices, rather than reimplementing them. For example, our software relies heavily on computational implementations of dMRI methods in the DIPY software [15] and visualization tools implemented in FURY [16]. Wherever possible, we use machine learning tools implemented in Scikit Learn [17], TensorFlow [18] or pyTorch [19]. We rely on the Brain Imaging Data Structure [20] as an approach to data management, and integrate

closely with the QSIPrep platform for preprocessing and for deployment of the pipelines we develop [21]. Though some of the tools have been used in previous publications [14,21–26], the present manuscript highlights the development of the ecosystem as a whole, and particularly several new developments that have been included since these publications. This includes new machine learning methods, new methods for high-performance computing, new interfaces for extensibility of the software, and improved interoperability. We provide computational notebooks highlighting example use-cases of the novel developments. These are available for download at https://github.com/tractometry/tractometry-ecosystem, and can be run interactively using Code Ocean at https://doi.org/10.24433/CO.1278808.v2 or Neurolibre [27,28] at https://preprint.neurolibre.org/10.55458/neurolibre.00037/. Overall, the software ecosystem introduced here provides an end-to-end solution for tractometry: it starts with raw dMRI data, and concludes with publication-ready visualizations and inferences about a range of scientific questions.

## Design and implementation

### Tractometry pipeline

The pyAFQ software (https://tractometry.org/pyAFQ) provides a configurable framework, allowing users to define specific methods and parameters for each stage of the tractometry analysis pipeline. pyAFQ was initially introduced in [14]. Here we introduce version 2.0, which is a substantial re-write of the core of software. This re-write focused on improving readability, configurability, and efficiency, as opposed to changes to the overall approach.

As a summary, we describe the pipeline as it would run under default settings. To start, we fit either Diffusion Tensor Imaging (DTI) [29,30] for single-shell or Diffusion Kurtosis Imaging (DKI) [31,32] for multi-shell data. We used these models to obtain the diffusion-derived scalar metrics fractional anisotropy (FA), mean diffusivity (MD), and, in the case of diffusion kurtosis imaging, mean kurtosis (MK). Unless a white matter mask is provided, we use FA thresholded of 0.2 to obtain a white matter mask, which we use as the seed and stop mask in tractography. Additionally, we fit constrained spherical deconvolution (CSD) to obtain fiber orientation distribution functions for tractography [33,34]. We perform tractography using methods implemented in DIPY. Subsequently, the T1-weighted Montreal Neurological Institute (MNI) template is non-linearly registered to the anisotropic power map (APM) of the CSD fit [35] using the symmetric normalization approach that is implemented in DIPY [36].

The registration step provides a transformation that maps canonical anatomical landmarks, probability maps, or other templates from MNI space to the subject's native space. The next phase is tract recognition, where each tractography streamline is classified as belonging to a specific tract based on the mapped templates or other criteria, as described in Adding new tract definitions.

The final step is to extract the tract profiles. Each streamline is resampled to a fixed number of points (100, per default), and the interpolated value of each diffusion-derived scalar metric (e.g., FA, MD) is calculated at each node. A mean profile is calculated by weighting the contribution of each resampled streamline, based on how similar the trajectory of this streamline is to the median position of the streamlines in the identified tract.

**Integration with QSIPrep.** QSIPrep outputs data in BIDS format, which can be directly input to pyAFQ using the GroupAFQ class with `preproc_pipeline` set to "qsiprep". Alternatively, pyAFQ can be run directly from QSIRecon as a reconstruction pipeline. QSIRecon was recently separated from the rest of QSIprep and builds post-processing workflows. There are two QSIRecon workflows that use pyAFQ: `pyafq_tractometry` and `mrtrix_multishell_msmt_pyafq_tractometry`. The first runs the default pyAFQ

pipeline, which can be customized using its associated JSON configuration file. The second runs MRtrix3's iFOD2 tractography using fiber orientation distribution functions from multi-shell multi-tissue constrained spherical deconvolution [37], and then imports that into pyAFQ for tract identification and tract profile calculation.

## Extending software functionality

A large number of configuration parameters provide detailed control over data modeling, registration, tract identification, tractography, and visualization for tractometry. Data modeling parameters allow the user to choose from numerous models implemented in DIPY, and then choose how their derivatives are used in subsequent tractometry steps (i.e., as tissue properties for tract profiles, as masks for tractography, as a contrast for registration, etc.). The registration parameters allow users to specify which of the suite of registration tools implemented in DIPY to use for registration, and set their parameters, while supporting various registration templates which are automatically downloaded, or subject contrasts from the data. Tractography parameters allow users to choose how to seed the tractography, whether to use deterministic or probabilistic tractography, and to set tractography parameters such as the maximum turning angle or maximal and minimal length. Tract recognition parameters are largely set in a `BundleDict` object as described below in Adding new tract definitions. Finally, visualization parameters allow users control of rendering parameters, including streamline resampling, shading bounds, opacity settings, and the choice of visualization backend (e.g., FURY, Plotly). These configurations provide flexibility for tailoring the pipeline to specific research goals.

**Tissue properties from other pipelines.** By default, pyAFQ computes tract profiles for FA and MD using DTI for single-shell data or DKI for multi-shell data. There are built-in options for using other tissue properties that are already implemented in DIPY. Additionally, we allow users to input tissue properties (or any other 3D image with a single numerical value per voxel) from other software. If the tissue properties are stored in the BIDS format, BIDS filters can be passed to automatically identify these images for each subject. Alternatively, the user can directly provide the paths to these images.

**Adding new tract definitions.** pyAFQ allows adding custom tract definitions directly through the software API using a `BundleDict` class instance. This class is built around a Python dictionary and specifies the tracts to process and how they are defined. Each key in the dictionary corresponds to the name of a white matter tract, and its value is another dictionary containing the parameters that define the tract's recognition criteria. Each criterion is optional. Many use regions of interest (ROIs) in the standard format of Nibabel's Nifti image files [38]. Here are the currently implemented criteria:

- **include**: A list of paths to Nifti Image files containing inclusion ROI(s). A streamline must pass through an inclusion ROI to be selected.
- **exclude**: A list of paths to Nifti files containing exclusion ROI(s). A streamline must avoid all exclusion ROIs to be selected.
- **start**: A path to a Nifti file containing the start ROI. A streamline must start in this region to be accepted.
- **end**: A path to a Nifti file containing the end ROI. A streamline must end in this region to be accepted.
- **cross_midline**: Boolean describing whether streamlines are required to cross the midline or are prohibited from crossing the midline to be accepted.
- **prob_map**: Path to a Nifti file which contains a probability map. Streamlines must have probability above a certain threshold to be accepted.

- **mahal**: Dictionary describing the parameters for "cleaning" the tract, as described in Yeatman et al. 2012 [7]. This involves removing streamlines that are unusually long or far away from the other streamlines in this tract, according to the Mahalanobis distance. By default, the same cleaning parameters are used as in Kruper et al. 2021 [14].
- **recobundles**: Dictionary which should contain a **sl** key and a **centroid** key. The **sl** key should be the reference streamline, and the **centroid** key should be the centroid threshold for the Recobundles algorithm, which uses streamline-based models as shape priors to select streamlines [39]. It can be used on its own or in tandem with ROIs.
- **qb_thresh**: Float which is the threshold, in millimeters, for Quickbundles cleaning [40]. This is a clustering-based algorithm which can complement the Mahalanobis-based cleaning, particularly when streamline length or distance distributions are non-Gaussian.
- **primary_axis, primary_axis_percentage**: String and integer indicating the primary axis the streamline should travel in and what fraction of a streamline's movement should be in the primary axis. This is useful for defining the vertical occipital fasciculus, but could be useful for other tracts. Can be one of: 'L/R', 'P/A', 'I/S'.
- **length**: Dictionary containing **min_len** and **max_len** defining the length range (in millimeters) for a streamline to be selected.

ROIs can be given in either the individual subject's space or the template space. The inclusion and exclusion ROIs can also be given tolerances in millimeters (i.e., distance from the ROI that a streamline can travel and still be included/excluded). Note that all of these criteria are optional and no default tract currently uses all of them at once. Currently, the criteria are used in the following order: (1) `prob_map`, (2) `cross_midline`, (3) `start`, (4) `end`, (5) `length`, (6) `primary_axis`, (7) `include`, (8) `exclude`, (9) `recobundles`, (10) `qb_thresh`, and (11) `mahal`.

If a streamline passes all steps for multiple tracts, an informative warning is displayed and the tie goes to whichever tract is first in the bundle dictionary.

**Code examples: Customizing the bundle definition.** Here is an example custom dictionary for adding a bundle definition for the acoustic radiation to pyAFQ using the automated anatomical atlas (AAL) [41]. First, we import pyAFQ's bundle dictionary module:

```
import AFQ.api.bundle_dict as abd
```

Then, you must provide your ROIs as Nifti images which will be provided to the pyAFQ bundle dictionary. Here, for the example, we assume there is a method to grab them:

```
AAL_roi = get_AAL_templates()
```

Finally, we have the syntax for providing the ROIs to pyAFQ. Here we provide them as start and endpoint ROIs, such that the resulting tracts and tract profiles will be oriented from start to end. Additionally, we specify that the tract should not cross the brain's midline.

```
ar_bundles = abd.BundleDict({
    "Left Acoustic Radiation": {
        "start": AAL_roi["AAL_Thal_L"],
        "end": AAL_roi["AAL_TempSup_L"],
        "cross_midline": False
    },
    "Right Acoustic Radiation": {
        "start": AAL_roi["AAL_Thal_R"],
```

```
        ”end”: AAL_roi[”AAL_TempSup_R”],
        ”cross_midline”: False
    }
})
```

Conversely, if you would like to add a Recobundles criterion and Quickbundles cleaning [42], you can get reference bundles from the Yeh atlas with this method from pyAFQ [42]:

```
import AFQ.data.fetch as afd
reco_ref = afd.read_hcp_atlas(80, as_file=True)
```

In this setup, the bundle specification includes the additional "recobundles" parameter, and the "qb_thresh" parameter specifying a final Quickbundles cleaning step [40].

```
ar_bundles = abd.BundleDict({
    ”Left Acoustic Radiation”: {
        ”start”: AAL_roi[”AAL_Thal_L”],
        ”end”: AAL_roi[”AAL_TempSup_L”],
        ”cross_midline”: False,
        ”recobundles”: reco_ref[”AR_L”][”recobundles”],
        ”qb_thresh”: 6
    },
    ”Right Acoustic Radiation”: {
        ”start”: AAL_roi[”AAL_Thal_R”],
        ”end”: AAL_roi[”AAL_TempSup_R”],
        ”cross_midline”: False,
        ”recobundles”: reco_ref[”AR_R”][”recobundles”],
        ”qb_thresh”: 6
    }
})
```

And here is how one would combine this with our default bundles for a run of pyAFQ:

```
ar_bundles = abd.default18_bd() + ar_bundles
```

## Statistical methods and machine learning

**AFQ-Insight for statistical modeling.** One common approach to modeling the relationships between tract profiles and individual differences is by fitting a linear model or linear mixed-effects model at every point in the tract profile and assessing the probability of the data given a null hypothesis (the p value) at each point independently. Because there are many points in each tract profile, this process is repeated many times, and it is susceptible to inflation of type 1 errors. Thus, it is appropriate to correct for the multiplicity of points along each tract profile, e.g., by using methods that control the False Discovery Rate (FDR) [43]. To implement these approaches, we rely on the statsmodels Python library to fit parametric models on a point-by-point basis. We implemented a wrapper and object-oriented interface to the functionality of this library that allows specifying the model to be tested and whether to apply FDR correction. To characterize statistical modeling with AFQ-Insight, we analyzed a previously published dataset in which 24 individuals with amyotrophic lateral sclerosis (ALS) and 24 matched controls were compared [44]. The data has previously been described; briefly:

a 3T General Electric scanner was used dMRI was acquired with 2mm isotropic voxels, 27 directions at a b-value of 1,000 $s/mm^2$. Data was processed as described in [44] and we used the tract profile produced there. For the original data collection all participants gave written informed consent, which was approved by the Ethics Committee of the University "Magna Graecia" of Catanzaro as described in [44]. The University of Washington IRB confirmed that reanalysis of anonynmized data is not human subjects research under its rules. For statistical analysis, we fit the model:

```
FA ~ C(Group)
```

in each of the 100 nodes along the right cortical-spinal tract, where the model is written out in the statsmodels syntax, which is very similar to the standard R model syntax, except that `C()` indicates a categorical/factor variable. FDR correction for multiple correction was applied [43]. This code is implemented as open-source software in https://tractometry.org/ AFQ-Insight.

**Tractable for generalized additive models.** Generalized additive models (GAMs) are an extension of the generalized linear model [45]:

$$g(y) = f_1(x_1) + f_2(x_2) + ... + f_n(x_n) + \epsilon, \tag{1}$$

where $g$ is a link function, $x_i$ is each of the covariates (e.g., age, spatial location along the tract) and $f_i$ are non-linear functions of these covariates. These functions take the form:

$$f(x) = \sum_{j=1}^{k} \beta_j b_j(x), \tag{2}$$

where $b_k(x)$ are smooth non-linear functions of the covariate $x$ (such as thin-plate splines [46]). The complexity of the model is determined by the number of smooth functions used ($k$), as well as by an additional penalty on the second derivative of the splines, which is weighted via a hyperparameter $\lambda$. Together, these hyper-parameters ($k, \lambda$) determine how expressive the model is allowed to be and how prone it is to overfitting. A flexible and powerful implementation of GAMs is offered by the mgcv library, implemented in the R statistical programming language [47]. To facilitate use of mgcv with tract profile data, we implemented an R software library called Tractable, which accepts as input the comma-separated value (CSV) output of pyAFQ and a light-weight model specification, and can then run models either for specific selected tracts or for each of the tracts that are present in the CSV. Tractable automatically determines the lowest value of $k$ that well accounts for the data using restricted maximum likelihood (REML) [48]. One of the concerns of GAM models for data with serial dependency and correlated noise, is that these autocorrelations can inflate type 1 errors. Therefore, data needs to be "pre-whitened" with an autoregressive model [49,50]. We rely on the `itsadug` software for an automated procedure to determine the degree of correction, and an AR1 model implemented in mgcv. For comparison of ALS and controls, we fit the following model (here in the R formula syntax):

```
fa ~ age + group + s(nodeID, by = group, bs = "fs", k = 9) + s
    (subjectID, bs = "re")
```

where the first term accounts for age effects; the second term tests for group differences; the third term is the combination of smooth functions that accounts for variation over the length

of the tract (and $k = 9$ was automatically determined from the data); the last term indicates that subjects are entered as random effects [51]. This analysis is implemented as open-source software in https://tractometry.org/tractable.

**AFQ-Insight machine learning methods.** We implemented a series of different models using the Tensorflow framework for deep learning [18], and results herein are based on this implementation. All of these models were implemented using the Keras API [52] of TensorFlow 2. Both the neural network code and the data augmentation code, discussed below, were inspired heavily by the implementations of Iwana and Uchida [53], with some modifications for use on tractometry data and an adjustment to the output layer activation functions for use in regression. As the PyTorch framework is gaining in popularity [19], we have also implemented these models in Pytorch as part of the AFQ-Insight software library and test them against the Keras implementation as reference. Open-source software implementing these methods is available at https://tractometry.org/AFQ-Insight.

**Data collection and data processing.** We used data from the Healthy Brain Network [54] that we previously processed and automatically quality controlled [55]. The measurements and processing are described extensively elsewhere [54,55]. Briefly, diffusion-weighted magnetic resonance imaging data was acquired with a spatial resolution of 1.8 x 1.8 x 1.8 mm$^3$; 64 diffusion directions were measured with b=1,000 s/mm$^2$ and b=2,000s/mm$^2$ (in some rare cases, b=1,500 s/mm$^2$, and b=3,000 s/mm$^2$ were used). The data were curated using CuBIDS and then processed using QSIprep [21]. Quality control of the data was automated using a deep learning algorithm that was trained on community-scientist inputs, achieving high accuracy on a "gold standard" subset that was examined in detail by a group of experts [55,56]. The original sample includes 2,747 subjects, aged 5-21 years, but after exclusion based on suitability for analysis and after selecting participants for which quality control was above 0, we included data from 1,817 participants in the present study. As described in [54], the data was collected with approval by the Chesapeake Institutional Review Board (https://www. chesapeakeirb.com/). Prior to conducting the research, written informed consent was obtained from participants ages 18 or older, or from their legal guardians, with assent obtained from participants younger than 18. The University of Washington IRB confirmed that reanalysis of anonymized data is not human subjects research under its rules. As previously described, we used pyAFQ to extract tract profiles of 24 major tracts in every subject (See S1 Fig). Each tract was divided into 100 equidistant nodes, and dMRI-derived tissue properties were projected into each of these nodes. Tissue properties of the different tracts were characterized using the Diffusional Kurtosis Imaging model implemented in DIPY [15,32]. To model the relationship between dMRI and age, we used fractional anisotropy (FA), mean diffusivity (MD) and mean kurtosis (MK) based on this model. Data were organized into a table in which each subject was a row, and each column represented one of these tissue properties in a node along the length of one of the 24 tracts. Data were harmonized across the different sites of the HBN study during training of the models using the COMBAT algorithm [57–59] implemented in the `neurocombat_sklearn` software library (https://github.com/Warvito/neurocombat_sklearn), which we have also integrated into AFQ-Insight.

**Regularized regression methods.** In previous work, we described the use of regularized linear regression models for machine learning with tractometry data [22]. Based on this work, we used a PCR Lasso algorithm as the baseline for comparison with non-linear models. In the PCR algorithm the data are first re-represented using a principal components analysis, and then the Lasso regularization algorithm is used to select the most informative principal components [60]. An internal loop of three-fold cross-validation was used to determine the level of regularization applied in the Lasso phase of this model.

**Neural network models.** A variety of neural network architectures were implemented. Neural networks can take advantage of the spatial adjacency of measurements. Therefore, we encoded the spatially contiguous tract profile data as the "length" dimension in these one-dimensional networks (in analogy to different time-points in a time-series, or different pixels in an image), and we encoded different tracts/metrics as separate channels (analogous to the R,G, and B channels for color in computer vision networks; see S1 Fig). Inspired by the use of neural networks in time-series classification, we borrowed heavily from the work of Iwana and Uchida [53], who evaluated different data augmentation methods using ten different time series classification networks. These models can be divided into three broad groups which we describe below: fully connected networks, convolutional neural networks, and recurrent neural networks.

- Fully connected network: This network is agnostic to the serial relationships between measurements, but it can integrate information in a non-linear manner, due to its multi-layer architecture. We use a fully connected multi-layer perceptron (MLP) model specifically tailored for time sequence labeling proposed by Wang et al. [61]. It ignores the serial structure of tract profile data by first flattening the input. It then passes the input through three hidden layers, each with 500 units. Dropout [62] is added after the input layer, with a rate of 0.1, and after each hidden layer, with a rate of 0.2. We refer to this network as mlp4.
- Convolutional neural networks: Convolutional neural networks incorporate information about spatial correlations by training a set of convolutional units based on the data. This makes convolutional networks particular sensitive to abrupt changes in the data (i.e., edges) and higher-order non-linear correlations between features of the data at different positions.
    - 1-D Visual Geometry Group: Iwana and Uchida [53] modified the original VGG [63] network to accept one-dimensional time series input. They altered the number of convolutional blocks and max pooling blocks depending on the length of the input in order to prevent excessive pooling. We refer to this network as vgg.
    - LeNet-5: This 1-D CNN is an adaptation of LeCun et al.'s [64] seminal network developed for handwritten character recognition. Similar to vgg, Iwana and Uchida [53] altered the number of convolutional blocks depending on the length of the input to prevent excessive pooling. We refer to this network as lenet.
    - 1-D Residual Network: Here, we adopt the one-dimensional ResNet of Fawaz et al. [65], who modified the original ResNet [66] to contain only three residual blocks with varying filter lengths and no max pooling. We refer to this network as resnet.
- Recurrent neural networks Recurrent neural networks are particularly well-suited for analysis of data with serial dependencies, as the network is designed to combine information about any position with its memory of nearby positions.
    - Long short-term memory (LSTM): This variant of RNN is designed to retain information about relatively long sequences of data. We adopt three variants of the LSTM model. The first (lstm1v0) is an application of the original LSTM proposal [67] consisting of a single LSTM layer with 512 units. The next two are adopted from a survey of LSTM hyperparameters for sequence labeling tasks [68]. These models have either one (lstm1) or two (lstm2) stacked LSTM layers, each with 100 units.
    - Bidirectional long short-term memory (BLSTM): these variants of the LSTM use both forward and backward recurrent connections. In the context of time series

analysis, one can think of these connections as enabling prediction and retro-diction, respectively. Here, they symmetrically consider both directions of the tract profile. We adopt two versions of the BLSTM, blstm1 and blstm2, which are identical to lstm1 and lstm2, except that they use bidirectional layers.

– Long short-term memory fully convolutional network (LSTM-FCN): this model is a multi-stream neural network developed by Karim et al. [69] combining an RNN and a CNN. The CNN stream consists of three 1-D convolutional layers with batch normalization, followed by global average pooling. The RNN stream consists of a single LSTM layer with 128 units and a dropout rate of 0.8. The outputs of these streams are then concatenated and fed into one fully connected node. We refer to this model as lstmfcn.

**Data augmentation.**   We implemented a range of augmentation methods that are used in analysis of one-dimensional time-series [53]. In the experiments described here, we applied jitter, scaling, and time-warping. Each of these operations were parametrized by a single number, which was scaled to the range of values within each tract and metric (i.e, FA, MD and MK).

**Machine learning model evaluation.**   Each model was evaluated for accuracy in age prediction 10 times. In each trial of the experiment, data were split into training (80%) and testing (20%) sets. The test set was set aside, to be used only in evaluating the trained network at the conclusion of training. We used the same test set, regardless of the size of the training data. Within each trial, different experiments were conducted with each of these test samples, where we sub-sampled the training data down to samples of size: 100, 175, 350, 700, 1,000 or the full training sample size of 1,453 subjects. Random seeds were fixed such that the same 10 test sets and the same training subsets were used with each model, facilitating the comparison across models. In cases where features could not be evaluated (e.g., because of localized issues with the dMRI data), data was imputed with a median imputation strategy applied within each set (train/test) separately.

For the baseline PCR Lasso model, the degree of regularization is set using an internal 3-fold cross-validation procedure within each training set.

For the NN models, Keras callbacks functions are utilized to stop training when there is no further improvement in performance and to store trained model weight (i.e., `EarlyStopping` with `monitor='val_loss'`, `min_delta=0.001`, `model='min'`, `patience=100`; `ReduceOnPlateau` with `monitor='val_loss'`, `factor=0.5`, `patience=20`). The ADAM optimizer was used in all cases [70], and based on initial pilot experiments, we set slightly different learning rates for each NN architecture.

**Modeling training sample size effects.**   We fit the following exponential learning model to describe the effects of sample size on model performance:

$$R^2 = \alpha - (\alpha - \beta)e^{-(x-x_{min})/\kappa}, \tag{3}$$

where $x$ is the number of samples in the training set and $x_{min}$ is the smallest training set used (here, $x_{min} = 100$ in all cases), $\alpha$ is the highest $R^2$ observed in the data. Two parameters are fit using non-linear optimization: $\beta$ is an estimate of the performance that would be achieved at the smallest training sample, $\kappa$ is a parameter that quantifies the rate at which performance improves with additional data. Given the simple exponential form of this curve, this parameter corresponds to the number of subjects at which performance improves to 63% of the difference between $\alpha$ and $\beta$. The model was fit to the data using the Scipy software library's [71] `optimize.curve_fit` function. The code for executing the experiments conducted

in this section, as well as results stored in CSV format, are available in the following GitHub repository: https://github.com/tractometry/afq-deep-learning.

## Accelerated computational performance

**Parallelization of model fitting with Ray.**  Parallelization of model fitting was done within the DIPY software library. We used Ray's `remote` function to parallelize model fits across chunks of voxels. Data was split into chunks based on the available number of cores and each core performed the work in parallel. The array of model parameters was then concatenated and passed to other DIPY methods to compute derived quantities, such as fiber orientation distribution functions.

**Parallelization of tractography with Ray.**  Within the pyAFQ software, we used Ray's `actor` objects to generate *n* tractography objects, which each received a separate batch of seeds to work with. The number of chunks is determined by the number of available cores by default. When all tasks were completed, the separate objects were concatenated into one file. Data management aspects of this process are facilitated by functionality implemented in the TRX file format [72] and, therefore, it is not implemented with other file formats, where concatenation would be slower.

**GPU acceleration of tractography.**  The GPU implementation of tractography is written in CUDA C. For direction getting, there is a choice of standard probabilistic or bootstrapped tractography [73]. These two were written in CUDA C based on the Python reference implementation available in DIPY implemented as `BootDirectionGetter` and `ProbabilisticDirectionGetter`. For `BootDirectionGetter`, only the `CsaOdfModel` and `OpdtModel` are implemented, again reimplementing the DIPY code in CUDA C [74,75]. For `ProbabilisticDirectionGetter`, the code accepts any ODF fit by DIPY. In other words, the code is model agnostic, which also means that future models could still take advantage of GPU accelerated probabilistic tractography. By contrast, due to the nature of bootstrapped tractography, where models are re-fit during tracking, models need to be implemented individually in CUDA C. Both of these direction getters also use DIPY's `local_tracker` re-implemented in CUDA C.

GPU tractography can be installed using `pip`. It uses `scikit-build-core` and `pybind11`, which in turn require `cmake` for `pip` installation. Once installed, it can be called from Python. An example script is provided for reference in the repository: https://github.com/dipy/GPUStreamlines. A variety of parameters can be passed through this interface. The maximum angle, minimum signal, stop mask and threshold, step size, relative peak threshold, minimum separation angle, and random seed are passed through this interface. These parameters are functionally the same as in DIPY, so reference DIPY documentation for more information on these parameters. Additionally, the software can take advantage of multiple GPUs if they are available using the `ngpus` parameter. The software must be recompiled to change the maximum streamline length (currently 501 steps in each direction) and the maximum number of streamlines per seed (currently 10). GPU tractography is run in batches of seeds, limited by the GPU size. These batches can be either (1) returned as lists of arrays which can be cast to Nibabel's ArraySequence object or (2) saved directly as a series of TRK files. We provide code to merge these series of TRKs into one large TRK. We also provide code to save the lists of arrays to a TRX file as they are generated.

**Benchmark experiments.**  We benchmarked the software on the University of Washington High Performance Computing cluster, HYAK, which uses cores from either Cascade Lake or Ice Lake Intel servers. For CPU tests, we requested an allocation of 64GB of RAM through the Slurm queuing system, and, unless otherwise specified, 4 cores. We varied the

number of cores for benchmarking parallelization with Ray, but not the amount of memory. For GPU tests, we used 64GB of RAM and a single NVIDIA A40 GPU. We used Python Version 3.10, pyAFQ Version 1.3.3, GPUStreamlines Version 0.1 and DIPY Version 1.8.0. We used a HARDI acquisition with 2x2x2 mm3 isotropic voxels, 150 b=1,000 $s/mm^2$ volumes and 10 b0 volumes previously described [11]. This data was collected and shared as approved by the Stanford University Institutional Review Board, and is available at https://purl.stanford.edu/yx282xq2090.

## Results

### An end-to-end tractometry pipeline

A typical tractometry analysis starts with standard preprocessing steps, which include (i) denoising; (ii) removing artifacts due to Gibbs ringing; (iii) correcting for subject motion (iv) correcting for effects of eddy currents; and (v) correcting for geometric distortions of the images due to magnetic field changes in regions where magnetic susceptibility changes. A variety of different tools exist to perform these steps (e.g., [76,77]), and their products can all be used as inputs to our ecosystem. Raw data that is organized in the Brain Imaging Data Structure is amenable to analysis with the QSIPrep platform [21], which closely integrates with other elements of our ecosystem. The AFQ tractometry approach is implemented in the pyAFQ library (Fig 1, red-filled circle), which is the core of the ecosystem. It is implemented in the Python programming language, using the Pimms library for workflow management [78]. Per default, pyAFQ uses methods implemented in DIPY to perform computational tractography and microstructural modeling. By default, it uses constrained spherical deconvolution and probabilistic direction getting for tractography [33], and outputs mean diffusivity and fractional anisotropy tract profiles estimated using either diffusion tensor imaging or diffusion kurtosis imaging, depending on the number of non-zero b-values [32]. However,

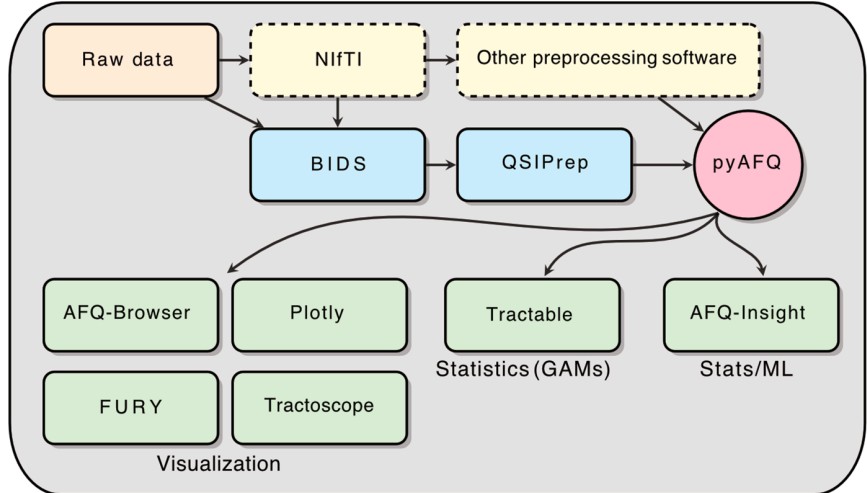

**Fig 1. The end-to-end tractometry pipeline beginning with raw dMRI data and proceeding through preprocessing (e.g., dcm2bids, QSIPrep, etc.) to pyAFQ-based tractometry.** pyAFQ accepts preprocessed data from BIDS, but does not require BIDS. Outputs can be visualized with tools such as AFQ-Browser, or integrated into machine learning workflows and statistical models. A typical tractometry pipeline flows from raw data, to the BIDS format, to QSIPrep, to pyAFQ. Then, we use the suite of tools shown in the green boxes on the results of pyAFQ. Descriptions for these software are available in Table 1.

it can also ingest tractography data and microstructural model maps that were computed in other software. It is designed to be highly extensible to accommodate a variety of use-cases (see section pyAFQ is highly extensible and interoperable). The primary output of pyAFQ are the tract profiles of tissue properties along all of the tracts that are identified by the software. They are stored in a CSV file in a long tidy format [79]. These tract profile data are the input to subsequent analysis steps. The Tractable library performs statistical analysis on these tract profiles (see section Statistical modeling). The AFQ-Insight library facilitates predictive machine learning analysis with tract profiles (see section Machine learning and predictive modeling). A variety of tools can be used to visualize the results, share them, and communicate about them (see section Extensible visualization tools covers all major use cases). Fig 1 shows how the tools interoperate and Table 1 gives a brief description of the tools.

## pyAFQ is highly extensible and interoperable

One of the primary goals of pyAFQ's software design is to provide users with the flexibility to extend it according to their specific needs. For instance, users can incorporate information on additional tracts, apply the software to different populations, compute or provide as input different maps of tissue properties. We showcase these extensions by way of practical examples.

**pyAFQ can be extended to include novel tract definitions.** In its default configuration, the pyAFQ software identifies 28 different tracts (Table 2), including a segmentation of the corpus callosum into 8 different sub-segments, that correspond to functional networks, based on work by Dougherty et al. [80]. To facilitate extensibility, the software was designed to be flexible to the addition of other tracts. The software implements a `BundleDict` object, which serves as a configuration for the definition of tracts. This object can take as its input

**Table 1. Descriptions of the different elements of the ecosystem and related software. Their connections are show in Fig 1. Many of these are hosted on https://tractometry.org/.**

| Name | Description | URL |
|---|---|---|
| QSIPrep | Configures pipelines for pre-processing dMRI data, such as: distortion correction, motion correction, denoising, etc. | qsiprep.readthedocs.io/ |
| pyAFQ | Automated Fiber Quantification (AFQ) in Python: performs tractometry on the preprocessed data, and generates tractograms for known tracts and their tissue properties in a CSV for visualization and statistics. By default, it also uses the interactive graphing library Plotly to generate HTMLs. Plotly is pip-installable and works in a headless environment, which makes it convenient on servers. | tractometry.org/pyAFQ/ |
| AFQ-Browser | A web-based visualization tool for exploring tractometry results from pyAFQ. pyAFQ will generate all necessary inputs to AFQ-Browser in the right format using the *assemble_AFQ_browser* method. | tractometry.org/ AFQ-Browser/ |
| FURY | A Python library for advanced 3D visualization, with specific methods for neuroimaging and tractography. This can be useful for visualizing the tractography files generated by pyAFQ. | fury.gl/ |
| Tractoscope | A tool for interactive visualization and analysis of tractography data designed to work specifically with qsiprep/pyAFQ-produced datasets. It uses the niivue library. | nrdg.github.io/tractoscope/ |
| Tractable | An R library for fitting Generalized Additive Models (GAMs) on the tractometry data output by pyAFQ in CSVs. | tractometry.org/tractable/ |
| AFQ Insight | A library for statistical analysis of tractometry data in python, including both machine learning and statistical workflows. It uses the CSVs of tissue properties outputted by pyAFQ. | tractometry.org/AFQ-Insight |

**Table 2. List of white matter tracts built-in to pyAFQ. Custom definitions can also be provided to pyAFQ. The Reference column shows the paper used to create the definitions, and the Set column shows how these are grouped in the software. However, these groups can also be combined and modified.**

| Tract Name | Color | Reference | Set |
|---|---|---|---|
| Left Anterior Thalamic | Blue | Yeatman et al. 2012 [7] | Default |
| Right Anterior Thalamic | Light Blue | | |
| Left Cingulum Cingulate | Green | | |
| Right Cingulum Cingulate | Light Green | | |
| Left Corticospinal | Orange | | |
| Right Corticospinal | Light Orange | | |
| Left Inferior Fronto-occipital | Brown | | |
| Right Inferior Fronto-occipital | Light Brown | | |
| Left Inferior Longitudinal | Pink | | |
| Right Inferior Longitudinal | Light Pink | | |
| Left Superior Longitudinal | Grey | | |
| Right Superior Longitudinal | Light Grey | | |
| Left Arcuate | Teal | | |
| Right Arcuate | Light Teal | | |
| Left Uncinate | Yellow | | |
| Right Uncinate | Light Yellow | | |
| Left Posterior Arcuate | Red | | |
| Right Posterior Arcuate | Light Red | | |
| Left Vertical Occipital | Gold | | |
| Right Vertical Occipital | Light Gold | | |
| Callosum Anterior Frontal | Dark Green | Dougherty et al. 2007 [80] | |
| Callosum Motor | Sky Blue | | |
| Callosum Occipital | Purple | | |
| Callosum Orbital | Dark Purple | | |
| Callosum Posterior Parietal | Rose | | |
| Callosum Superior Frontal | Sea Green | | |
| Callosum Superior Parietal | Pale Yellow | | |
| Callosum Temporal | Dark Red | | |
| Left Inferior Cerebellar Peduncle | | Jossinger et al. 2022 [83] | Cerebellar |
| Right Inferior Cerebellar Peduncle | | | |
| Left Middle Cerebellar Peduncle | | | |
| Right Middle Cerebellar Peduncle | | | |
| Left Superior Cerebellar Peduncle | | | |
| Right Superior Cerebellar Peduncle | | | |
| Left Acoustic Radiation | | Tzourio-Mazoyer et al. 2002 [41] | Acoustic Radiations |
| Right Acoustic Radiation | | | |
| Left Optic Radiation | | Caffarra et al. 2021 [84] | Optic Radiations |
| Right Optic Radiation | | | |
| Left Superior Longitudinal I | | Sagi et al. 2024 [85] | Superior Longitudinal Sub-divisions |
| Right Superior Longitudinal I | | | |
| Left Superior Longitudinal II | | | |
| Right Superior Longitudinal II | | | |
| Left Superior Longitudinal III | | | |
| Right Superior Longitudinal III | | | |
| Left Optic Tract | | Kruper et al. 2023 [82] | Posterior Retino-geniculate Visual Pathway |
| Right Optic Tract | | | |
| Left Posterior Optic Nerve | | | |
| Right Posterior Optic Nerve | | | |

waypoint regions of interest (inclusion and/or exclusion), as well as probabilistic maps [81], endpoint regions of interest and specific shape criteria [82]. Users can add tracts by including code that defines this tract based on a combination of these criteria. Criteria can be defined from an atlas in template space, and mapped to each subject space, or be defined as BIDS filters or paths for each subject, allowing use of functionally-defined regions of interest that may vary in their 3D position from subject to subject. Examples of pathways that were previously added to the software are presented in Table 2.

**pyAFQ extends to different populations.** pyAFQ works with a range of different populations. For example, newborn infants' brains differ substantially from those of children and adults: they differ in size and volume, as well as in the curvature of several of the major pathways [86]. Thus, analyzing data from infant datasets requires substantially different definitions of the tracts. Based on previous work to translate the AFQ methods to infant data [25], users of pyAFQ can incorporate infant-specific templates [87] and infant-specific waypoint regions of interest that take into account these differences to accurately delineate the tracts (Fig 2A and 2B). In a previously published study [26], we demonstrated that these methods produce very accurate tract delineations, as validated in comparison to tracts that were manually delineated by experts.

**pyAFQ interoperates with other tract identification methods.** Recobundles is an alternative method for recognizing tracts by matching clusters of streamlines to a predefined streamline-based atlas [39]. The DIPY implementation of Recobundles is integrated into pyAFQ as another option for the definition of tracts. The pyAFQ documentation provides an example that uses Recobundles in conjunction with an atlas of eighty white matter tracts [42] (A subset of these is shown in Fig 2C, D). Additionally, individual bundle definitions can be customized to allow the Recobundles algorithm [39] to be used in conjunction with all of pyAFQ's other recognition methods, such as probability maps and regions of interest. Users of other tract delineation methods, such as TractSeg [88], WMQL [89], and deepWMA [90], or single tract-specific delineation methods such as RGVPSeg [91], can use the AFQ tract profile calculation implemented in DIPY directly on the outputs of these methods.

## Extensible visualization tools covers all major use cases

Data visualization serves a variety of roles in the scientific process: as a tool for quality control, for data exploration, as well as for communicating about the results. With these various roles in mind, the AFQ ecosystem supports a plethora of visualization outputs. The pyAFQ pipeline produces an html report for each participant in a dataset that can be used to explore subject-level results interactively in a web-browser. This visualization is based on the three-dimensional visualization capabilities of the Plotly software library and can be used as the basis for quality control of large datasets, because it is easy to share through standard web technologies. An example of this capability is available at https://tractometry.org/pyafq_qc_demo/. The outputs of the pipeline are also compatible with our previously-developed AFQ-Browser [23] (Fig 3C). This is a tool that enables exploratory data analysis of group-level data, by enabling filtering and sorting of subject data and grouping of subjects based on tract profiles or other characteristics (e.g., age, sex). Publication quality visualizations are easily obtained with the FURY library [16] (Fig 3A, also all the images in Fig 2). FURY is a flexible framework for generating high-quality 3D renderings and interactive visualizations. It currently uses The Visualization Toolkit (VTK) [92] and future versions will rely on pygfx/webGPU. The outputs of pyAFQ are compatible with FURY, and pyAFQ can optionally generate FURY visualizations of its output tracts on systems where FURY is installed. Finally, pyAFQ outputs can be input to Tractoscope, a next-generation browser-based visualization tool built

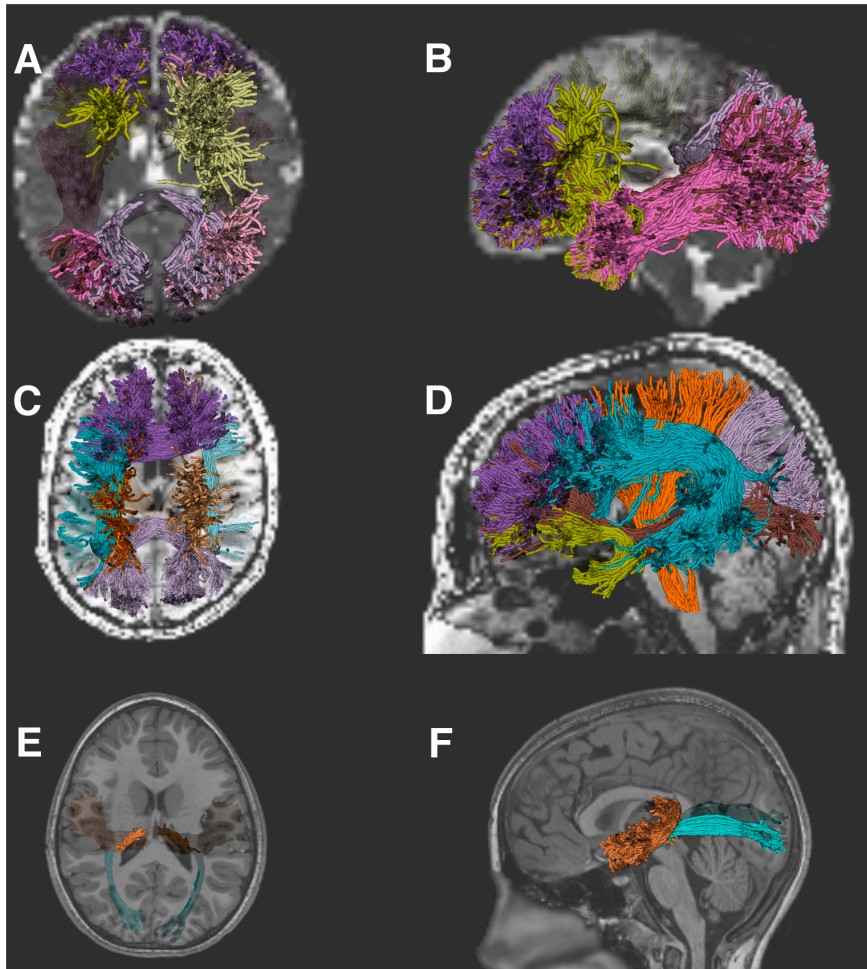

**Fig 2. Extending the pyAFQ software.** Panels A and B show select tracts recognized using Baby pyAFQ, with MRtrix3 for tractography [37], in an example baby subject. Panels C and D show select tracts recognized using Recobundles [39] in the Stanford HARDI subject [11]. Tracts in panels A-D were selected for visual clarity. Panels E and F show the acoustic and optic radiations recognized in subject NDARAA948VFH from the Healthy Brain Network Processed Open Diffusion Derivatives dataset [54,55]. All of these recognized tracts are results of Python scripts from pyAFQ's examples library, which demonstrates pyAFQ's extensibility. Panels A, C, and E show the axial plane from above and panels B, D, and F show the sagittal plane from the left.

on niivue [24,93] (Fig 3D). Tractoscope currently relies on accessing tractometry results that are openly available on Amazon's simple storage service (S3). We have previously shared tractometry results from the Human Connectome Project (HCP) and the Healthy Brain Network (HBN) on S3 through the FCP/INDI [94] and the Open Neuro Data [95] AWS Open Data buckets. These datasets can both be viewed on the instance of Tractoscope that is available at https://nrdg.github.io/tractoscope.

## Deriving insights from tractometry with statistics and machine learning

The output of pyAFQ is a table of tract tissue properties that are organized in a tidy long format [79]. This format is amenable to a variety of statistical and machine learning (ML) analysis methods. The purpose of statistical methods and ML applied to tract profile data

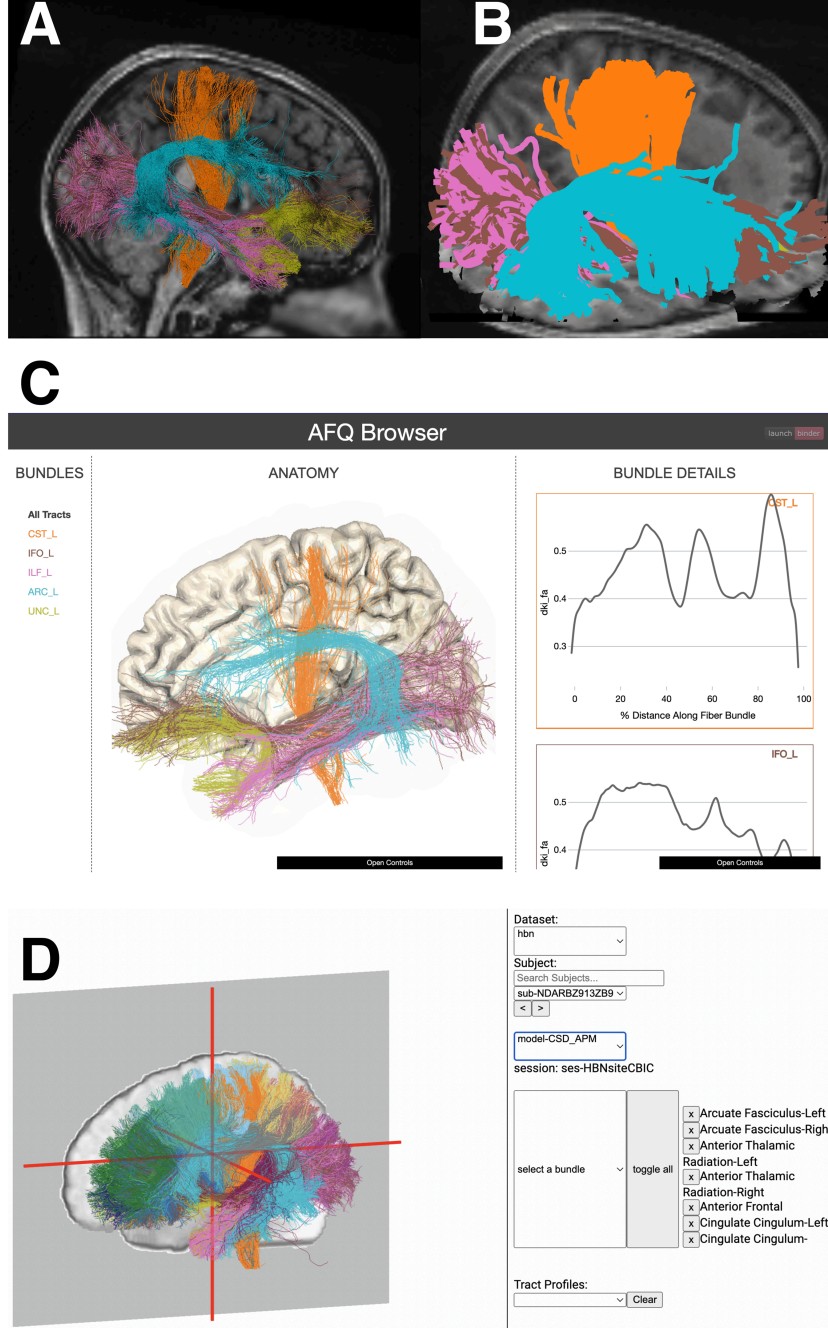

**Fig 3. Visualization tools available in the tractometry ecosystem.** All tools are visualizing subject NDARBZ913ZB9, randomly chosen from the HBN study [54,55]. **(A)** FURY visualization of five tracts (all in the left hemisphere): Corticospinal tract (orange), arcuate fasciculus (blue), inferior fronto-occipital tract (brown), uncinate fasciculus (yellow), and inferior longitudinal fasciculus (pink). **(B)** Plotly visualization of the same tracts, using the same color scheme, which can be rendered into an html webpage for quality control. **(C)** Visualization of this subject and tracts in AFQ Browser. The color scheme is defined on the left in the BUNDLES column, the anatomy is shown in the middle, and the subject's tract profiles are shown on the right. Tract names can be selected on the left column to highlight in on that tract's tissue properties and anatomy (a running example is also available at https://yeatmanlab.github.io/AFQBrowser-demo). Additionally, if multiple subjects are provided, AFQ Browser will show all of the tract profiles overlaid in the "Bundle Details" display on the right-hand side. **(D)** Web-based visualization of this subject and all their tracts with Tractoscope. Tractometry results of all HBN subjects are available to view at https://nrdg.github.io/tractoscope/.

is to understand the relationships that exist between individual variability in the tissue properties of white matter tracts and individual variability in other characteristics. For example, it can be used for comparing different groups of individuals (e.g., patients and controls) or modeling the relationship with a continuously varying characteristic (e.g., individual age). The AFQ-Insight software library supports both statistical modeling and ML approaches to these analysis questions using the results of pyAFQ. In addition, the Tractable software library supports a particular variant of statistical modeling that relies on generalized additive models [96].

**Statistical modeling.** We analyzed data from a previously-published study of a group of patients with amyotrophic lateral sclerosis (ALS) and matched control subjects [44] (Fig 4). One approach to this comparison would be to conduct a mass-univariate statistical analysis, to compare between the groups at each point along the tract profile. Indeed, when AFQ-Insight is used to conduct such a comparison, several points in the FA tract profile along the right corticospinal tract are found to be significantly different between the groups, using a threshold of $p<0.05$, FDR-corrected for multiple comparisons. However, this approach does not account for the statistical dependencies between neighboring points along the tract profile. A recently-proposed alternative is to use generalized additive models (GAMs) [45,51], which explicitly model the spatial contiguity of the tract profiles, and also allow to account for the auto-correlation of model residuals [49]. Tractable uses an automated procedure to determine the number of non-linear functions needed to represent the data, while controlling for over-fitting. In this case, $k = 9$ was determined to provide sufficient flexibility for the model based on this procedure. In addition, an AR1 process is used to model the auto-correlation, and an automatic procedure (based on itsadug functionality) is used to determine the

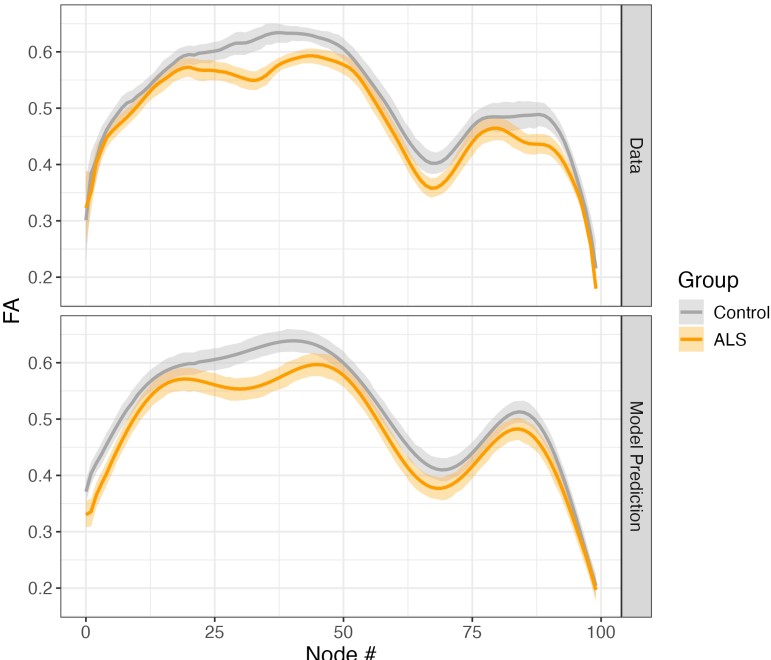

**Fig 4. The `Tractable` software was used to fit a GAM to corticospinal tract profiles of fractional anisotropy (FA) in participants with ALS (orange) and matched controls (gray).** The software includes functionality to visualize empirical data (top) and model fit (bottom).

magnitude of the AR1 process [49]. Even after accounting for autocorrelated errors, a statistically significant difference between groups was found ($p<0.05$). Furthermore, the model accurately fits the data (adjusted $R^2 = 0.56$; Fig 4 bottom panel).

**Machine learning and predictive modeling.** While statistical modeling approaches focus on testing the strength of parametric relationships or on null hypothesis testing (e.g., a null hypothesis that there are no differences in terms of white matter tract tissue properties between two groups), machine learning methods focus on building predictive models that integrate information over multiple points along a tract or along multiple tracts to accurately classify the group to which an individual belongs (in a classification task) or accurately model a continuous characteristic of the individuals, such as age (in a regression task). AFQ-Insight provides access to a range of machine learning algorithms that capitalize on the structure of tract profile data. It contains implementation of the sparse group lasso [22], which enables regularization on a within-tract and across-tract basis, as well as implementations of several other methods, including both regularized regression, principal-component regression (PCR) and functional PCA regression. In addition to these methods that are based on linear regression, AFQ-Insight includes neural network models implemented both in TensorFlow and pyTorch. To demonstrate the utility of these methods and assess their data requirements, we conducted evaluation experiments in the HBN dataset, comparing Lasso-regularized PCR as a baseline with neural network models on a standard benchmark task of age prediction.

To assess the impact of training set size, each NN algorithm and the baseline model were trained on a varying number of subjects. For each training set size, the models were each fit 10 times, allowing us to assess the reliability of model predictions. To facilitate comparison across models, the random seeds in each experiment were stored so that each model was trained on the same 10 different random samples of 80% of the subjects (n = 264), and the test sets used in each of these 10 iterations of training were identical across the different models.

**Baseline model.** We compared neural network performance to a baseline regularized linear regression algorithm: Principal Components Regression with Lasso regularization (PCR-Lasso) [22]. This algorithm starts by reducing the dimensionality of the data using principal components analysis (PCA) and then uses the Lasso regularization approach to select the PCs that best fit the linear relationship with age. This is a strong baseline, as we have previously found this algorithm to reach state of the art performance on this task [22]. Indeed, this algorithm reaches maximal performance at $R^2 = 0.62 \pm 0.03$ (SEM; Fig 5a). To characterize the dependence of algorithm performance on training set size, we fit an exponential learning curve model to these results (for example, in Fig 5a, the solid black line). The model uses the maximal performance observed in the data ($\alpha$), and then fits two additional parameters: $R^2$ at the smallest sample size ($\beta$) and the rate in which the curve between these two performance levels changes with increased training set size ($\kappa$). We find that PCR-Lasso improves substantially with the addition of training data. Performance in the smallest sample size (n=100) being $R^2 = 0.38 \pm 0.03$ (SEM). Training set size dependence is such that $\sim 63\%$ of the difference between the lowest and highest $R^2$ (captured in $\kappa$) is achieved with 413 subjects.

**Neural network performance.** As expected, NN algorithms are "data hungry": all the algorithms exhibit poor performance at small sample sizes, with none of the NN algorithms reaching even the modest performance of the baseline model at the smallest sample size (Fig 5). Nevertheless, we found that, for most of the NN algorithms, $R^2$ reaches an asymptote within the range of training set sizes used here, but maximal performance varies widely. The mlp4, resnet, and lstmfcn algorithms all reach no better than $R^2 = 0.5$, even when the largest training set is used (Fig 5B and 5C). On the other hand, some RNN algorithms reach parity with the baseline (blstm2; $R^2 = 0.62 \pm 0.016$ SEM, Fig 1d), or even slight improvement above

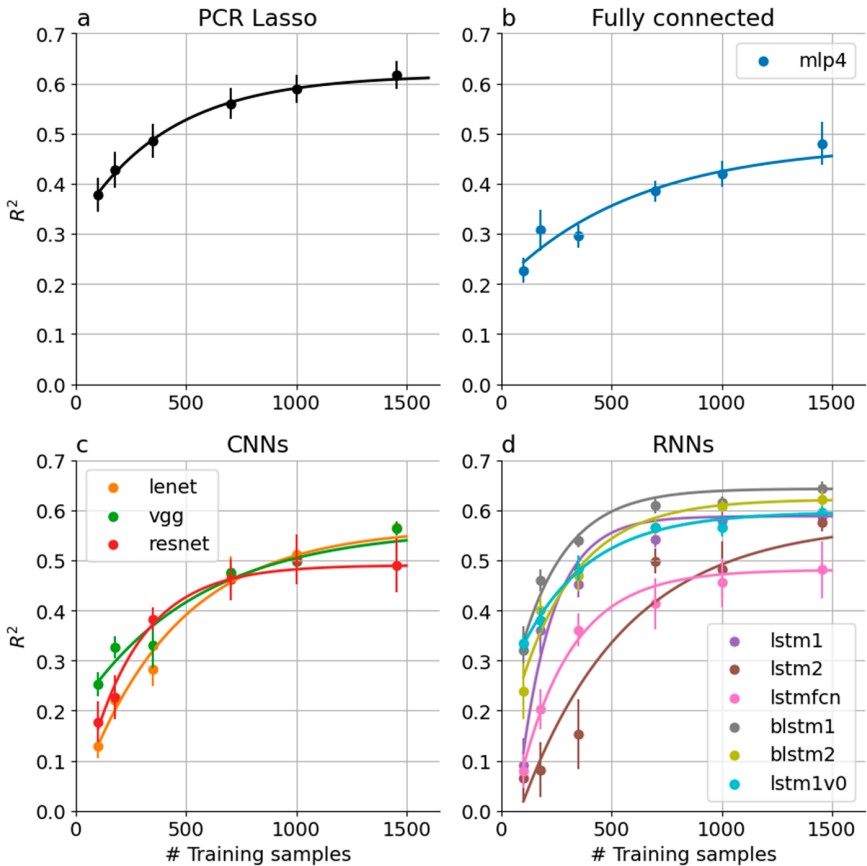

**Fig 5. Performance of all brain age models increases with the number of subjects in the training set.** (a) PCR Lasso, the linear baseline model starts at a relatively high $R^2$, even at the smallest sample used here, and then increases from there. (b) MLP4, a fully connected algorithm has a similar learning curve, but it starts at much lower $R^2$, (c) Convolutional neural networks (CNNs) start at an even lower $R^2$ with small sample sizes, but increase precipitously reaching higher levels of performance at large sample sizes. (d) Recurrent Neural Networks (RNNs) have a variety of different performance characteristics, but are also overall very data hungry.

the baseline (blstm1; $R^2 = 0.64 \pm 0.015$ SEM, Fig 1d). Importantly, some algorithms don't seem to reach asymptotic performance, even with the largest sample size that we used, and their performance may improve with even larger samples. This seems particularly true for the more modestly sized CNNs: lenet and vgg (Fig 5c), as well as the lstm2 RNN algorithm (Fig 5d). Data augmentation methods are very effective in improving NN model performance [97], and we found that augmenting the data with techniques borrowed from the time-series analysis literature [53] moderately improves performance in this task for some models (S2 Fig).

## Compute time accelerated through new computational paradigms

Due to its data-intensive and compute-intensive nature, tractometry is time-consuming. Running the default pyAFQ pipeline with 1 million seeds on a single subject data in a HARDI acquisition takes 1 hour, 14 minutes and 10 seconds $\pm 27$ seconds (SD over 10 runs). Of this, 58% of the time (43 minutes) is devoted to tractography, the process of generating candidate brain connection trajectories.

To speed up tractometry calculations we have advanced parallelization at two different levels. First, because most tractometry processing is done within individual subjects, there is an opportunity to parallelize computations across subjects. This makes processing multi-subject datasets an embarrassingly parallel process. In pyAFQ, this is implemented by relying on the pydra Python library to parallelize using concurrent futures (in the single-machine setting) or using the SLURM workload manager (in high-performance computing clusters) [98,99]. The API for between-subject parallelization mimics very closely the standard API used to process groups of subjects, requiring only a change to the name of the function that is used, making it easier to transition from prototyping on a small subset of subjects to large datasets with many thousands of subjects.

Secondly, within every subject, we added another layer of parallelization using the Ray distributed computing framework [100]. This operates at two different levels: within the DIPY software library, Ray is used to parallelize fitting of models of the diffusion signal within each voxel. This can be useful for popular models such as multi-shell multi-tissue constrained spherical deconvolution (MSMT-CSD), which uses convex optimization and is, therefore typically time-consuming: for the HARDI dataset, fitting MSMT-CSD to all the voxels takes 1 hour and 9 minutes (Fig 6A). This can be reduced to just under 3 minutes using Ray parallelization on 32 cores. In addition pyAFQ uses the Ray framework and the TRX file-format to parallelize tractography. For the default pyAFQ pipeline, using 1 million seed points, on 16 cores, ray reduces tractography time to 4 minutes and 5 seconds ±19 seconds (SD over 10 runs), an approximately 10x speedup. Also, pyAFQ uses GPU-accelerated tractography. While past versions of DIPY's GPU-accelerated tractography [101] only included specific algorithms, our recent developments enable multiple reconstruction algorithms and tractography approaches. The current version of the software can generate a probabilistic tractography from 1 million seeds in 37.7 seconds on a NVIDIA A40 GPU with the Stanford HARDI dataset (22.5 seconds on the GPU, 15.2 seconds for file writing), providing approximately 60x speedup relative to the baseline (Fig 6B.)

## Availability and future directions

The software ecosystem presented here provides a robust, efficient and extensible platform for reproducible end-to-end tractometry. Tractometry is broadly useful, well validated [14], and anatomically accurate [102]. It has been shown to be useful in a diverse set of contexts,

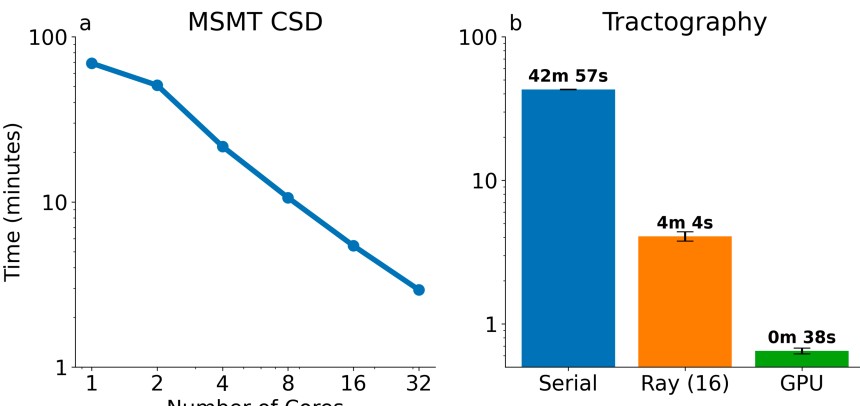

**Fig 6. A MSMT-CSD computation time using Ray versus the number of CPU cores provided.** With 32 cores, this takes just under 3 minutes. **B** DIPY tractography timing comparison across three methods: Serial (the DIPY default), Ray with 16 cores, and using the GPU accelerated version. Error bars show standard deviation across 10 trials.

from studying the visual system [103,104] to cerebral small vessel diseases [105,106], as well as neurological, neurodevelopmental, or psychiatric disorders such as autism spectrum disorder [107,108], traumatic brain injury [109,110], and catatonia [111]. Tractometry is useful for examining developmental and aging trajectories in the white matter [24,112–114] as well as exploring the relationship between brain tissue properties and individual cognitive or behavioral differences [115], and changes in these properties due to learning [116–118]. The present article introduces the next generation of tools that we have developed and used over multiple previous studies [14,21–26]. This article, in contrast, focuses on the novel improvements to the ecosystem and its integrated nature.

In the present article, we demonstrated that the tools that comprise the ecosystem that we have developed are interoperable, extensible, and efficient, allowing their application to the use-cases described above, and to others. A flexible framework for defining white matter tracts is implemented using a wide range of customizable criteria, allowing users to tailor tract recognition to their specific needs. By combining these custom definitions with the default tracts, users can seamlessly extend the software's capabilities to accommodate novel discoveries. A variety of visualization tools support quality control, data exploration and scientific communication, as well as data sharing.

In addition, we demonstrated a variety of ways in which this ecosystem supports deriving scientific insights from such data through its tools for statistical modeling and machine learning. These approaches provide complementary strengths. Statistical modeling focus on testing the strength of parametric relationships or on null hypothesis testing (e.g., a null hypothesis that there are no differences in terms of white matter tract tissue properties between two groups). In contrast, machine learning methods focus on building predictive models that integrate information over multiple points along a tract or along multiple tracts to accurately classify the group to which an individual belongs (in a classification task) or accurately predict a continuous characteristic of the individuals, such as age (in a regression task).

In the machine learning setting, the AFQ-Insight library provides several neural network models. While fully connected neural networks have previously been used in the analysis of tract profiles [119], we introduced a large collection of neural network models, including convolutional and recurrent neural network models enabling a multitude of modeling approaches. In comparing different machine learning approaches to tract profile modeling, we found that brain age modeling is more accurate with deep learning algorithms than with regularized linear regression models, although the advantage is not very large in the specific task of age prediction. Our results suggest that deep learning (DL) approaches should be used with caution, and with comparison to strong linear baseline models, such as the PCR Lasso model that we used here. This is precisely the kind of approach that is facilitated by the AFQ-Insight software. Moreover, as we posited in previous work [22], and consistent with previous findings [112,120,121] these results support the idea that brain development in childhood and adolescence is wide-spread throughout the brain and a process that occurs in tandem across large parts of the white matter. Under these conditions, non-linear and invariant modeling of the data may provide only minor benefits in terms of model accuracy. Nevertheless, RNNs and the blstm1 model, in particular, can emerge as superior to the linear baseline: this model has higher accuracy than the linear model at best, and its lowest $R^2$ performance is not much worse than the linear baseline, while it also requires less data than the linear baseline for similar performance. Presumably, this superior performance reflects the match of this particular model architecture to the data: it is most well-suited to represent the sequential serial dependencies in tract profile data, while considering these in any point in both directions (in contrast to the LSTM models, which consider only one direction of dependencies, more suited to time-series data). The improvement in $R^2$ does come at the cost of increased computational

time. In addition to these findings, CNNs, and particularly resnet and vgg, may gain even more in performance with increased sample sizes, as they do not seem to asymptote within the range of training sample sizes tested here, and they benefit from large amounts of augmentation. As we have previously demonstrated, DL approaches may be even more advantageous in other settings: In a study of a large (n>900) group of participants with glaucoma and tightly-matched controls, we found that the DL models similar to the resnet model benchmarked here can accurately classify glaucoma based on tractometry data from visual white matter pathways, while a linear baseline model (ridge-regularized logistic regression) cannot [24]. There are many additional applications of this approach, and we provide several additional examples in the documentation for the AFQ-Insight software library, as well as in the Code Ocean and the Neurolibre computational companions to this article [28].

While these models can be quite accurate, they can be opaque in terms of interpretation. The rich data provided in tract profiles also supports hypothesis-driven statistical analyses that are more readily interpretable, making it useful for both clinical and basic neuroscience research. By adhering to tidy data formats, our ecosystem allows interoperability between Python and R and supports future expansions to large-scale databases. One of the challenges of statistical modeling of individual differences in tract profiles is that data are sampled at an arbitrarily defined number of points along the tract and the data in neighboring points is not independent. In the machine learning setting, this is addressed by using regularization techniques, or neural network methods that account for these dependencies. In the setting of statistical modeling, Muncy et al. [51] proposed to address this by using generalized additive models [96], which model the overall shape of the tract profiles. These models provide an alternative to standard point-by-point or tract-averaged analysis that is currently common. This modeling approach is still nascent, but as the field evolves to improve this modeling approach, the software that we have developed will provide a basis to incorporate these advances. It is very useful to have both statistical modeling and predictive machine learning approaches so tightly interoperate, as these methods are complementary and can be fruitfully integrated [122,123]. In addition to these approaches, tract profile data generated through our ecosystem can be used in normative modeling [24,119,124], which is a target for future developments. To generate tract profiles in pyAFQ, we collapse tissue property estimates along the length of a recognized tract into a 1-dimensional series, and discretize it to some number of points, typically 100. While this approach has limitations, alternative methods such as FMQ, BUAN, and phybers offer diverse strategies [124–126]. By following tidy data format standards and prioritizing open-source software development, this ecosystem remains flexible and adaptable to such new advances.

Integration with QSIPrep [21] provides a turnkey solution for analyzing raw dMRI data. In this configuration, pyAFQ also complies with the specification for BIDS apps [127]. Together with new tools for data management [128], this makes it easier for users of our ecosystem to use a large range of methods matched to their datasets and questions and to deploy them in a reproducible and scalable manner on large openly available datasets, such as the Human Connectome Project (HCP), the Healthy Brain Network (HBN), Cam-CAN [3,129,130]. We have already published and made openly available tract profiles from HCP [115] and HBN [55]. All software provided here is version-controlled and publicly accessible, making it valuable for collaboration and ensuring reproducibility when performing tractometry on datasets with ongoing data collection, such as the Adolescent Brain Cognitive Development (ABCD) study [4,131] and Healthy Brain Cognitive Development (HBCD) study [132]. Another benefit of the QSIPrep platform is that enables mixing and matching various methods for analysis of dMRI data, including the many dMRI models that are implemented in DIPY but also the AMICO [133] implementation of the NODDI model [134], which is not currently directly

available through DIPY. This is also useful for tractography methods, where QSIPrep provides straightforward access to methods implemented in the popular MRtrix3 software [37]. The interoperability of these methods through a common interface should facilitate comparisons of different analysis pipelines, and use of the appropriate methods in each dataset.

To facilitate the use of the tools with large datasets users of pyAFQ can take advantage of parallelization, which is implemented both across-subject and within-subject, to achieve substantial speedup of computing. For within-subject parallelization, we demonstrated that Ray for Python enables efficient parallel execution of tasks that are highly independent [100]. This is a very common setting in neuroimaging, where parallelization can be performed across voxels or streamlines, and in many other scientific computing settings, to which these findings should generalize. While Python (and R) have historically been considered slower for compute-intensive tasks, this disadvantage has been mitigated through the development of optimized libraries and frameworks, such as Numpy [135], Scipy [136], and DIPY [15]. Our specialized ecosystem for tractometry builds upon a robust ecosystem of existing open-source software for scientific computing. For across-subject parallelization we have previously demonstrated that cloud computing can be used to significantly accelerate tractometry processing of datasets with thousands of subjects in a cost-effective manner [55,115,137]. However, in cases where researchers have access to a high-performance computing cluster, across-subject parallelization can also be achieved using standard queuing systems, such as SLURM [99]. We have now also implemented an application programming interface (API) that automatically parallelizes tractometry across subjects in settings in which SLURM is available, using the Pydra dataflow engine [98]. In addition, in settings in which GPUs are available, these can be used to additionally substantially speed up processing of each subject's data.

To further future-proof the use of these tools in a range of settings, we have incorporated the newly-proposed standard TRX file format, which provides a multitude of benefits, especially with large and high-resolution data [72]. Both Ray- and GPU-accelerated tractography algorithms use TRX to save arbitrarily large tractograms to a single file, as they are generated. In addition to the speedup provided, this alleviates memory issues that can arise when handling the large-scale tractography data. TRX allows for more flexibility in data types used for saving large tractographies, and has built-in compression, resulting in a better than 2-fold reduction in file size with little associated loss in information [115]. We anticipate that as the TRX file format itself improves, the use of this parallelization approach will have even larger benefits for users of our ecosystem.

Future developments in the ecosystem include extensions of the methods to data from young children. As demonstrated in pyAFQ is highly extensible and interoperable, the software already extends to neonate data. However, the brain develops very rapidly during early life and the methods are not expected to work well with children between 6 months old and approximately 5 years. After 5 years, adult-based anatomical constraints can be used. This is particularly important with the availability of data from studies such as HBCD that are conducting longitudinal dMRI measurements in children between age 0 and 2 years. Another future extension would be adapting AFQ for non-human data. This process would entail creating template ROIs and selecting a template image for registration, with the work on babyAFQ serving as a useful point of reference. Finally, standardization of tractometry outputs is an ongoing area for development. Although our software uses BIDS for inputs, its derivatives rely on advanced but as-yet unformalized drafts [138]. For now, they serve as a proof of concept to help advance formal standardization. To summarize, our ecosystem is efficient, adaptable, scalable, and is set to support reproducible, novel advancements in tractometry and neuroimaging research. The methods described here are all accessible as

open-source software freely available at https://tractometry.org. Detailed computational note-books demonstrating the various uses of the software are also available to download at https://github.com/tractometry/tractometry-ecosystem and can be run interactively through Code Ocean at https://doi.org/10.24433/CO.1278808.v2 and Neurolibre [27,28] at https://preprint.neurolibre.org/10.55458/neurolibre.00037/ .

## Acknowledgments

We would like to thank Howard Chiu and Audrey Luo for feedback during the development of the software described herein.

## Supporting information

**S1 Fig. Pipeline for extraction of parameters for ML and DL.** (A) Diffusion MRI (dMRI) data from the HBN-POD2 sample is used as input to the algorithm. (B) In the initial step, whole-brain tractography is conducted (C) Major white matter pathways are identified based on the trajectory of the streamlines. Here, the arcuate fasciculus (blue), cingulum cingulate (green) and corticospinal tract (orange) are displayed as an example. (D-F) Features of the tissue along the length of each tract are extracted from the image into a "tract profile". For each of the identified white matter pathways, the tract profile is extracted as a one-dimensional vector of numbers. (G) The brain age prediction problem set up as a linear model. Here $y$ is a column vector of the ages of the different subjects in the sample, $X$ is a matrix with each row containing the dMRI tract profiles of one subject. Features from each tract are color-coded as before. For example, all of the 100 values of the FA in the arcuate fasciculus may be stored in the columns indicated in blue. (H) To use as input to NN algorithms, the data in $X$ is transformed into a three-way tensor, with one dimension associated again with the subjects, another dimension associated with the position along the length of each tract (from 1 to 100) and the last dimension (known as the "channel" dimension in analysis of images) associated with the different tracts and metrics. (I) Schematic of a multi-layered neural network where the data from one subject (i.e., one row of the three-way tensor) is provided as input and the output is compared to the age of the subject ($y$) and the error can be propagated back to adjust the parameters of the network.
(PNG)

**S2 Fig. The effects of data augmentation on NN age prediction performance.** Data augmentation introduces random noise to each sample of the training data in each batch of training. This method can help NN algorithms with a large number of parameters generalize better, by preventing the memorization of the samples in the training set. When augmentation levels grow very large, however, the signal in the data is overwhelmed by the noise that is added in augmentation, and the algorithm can no longer learn. In our data, we found that augmentation can have dramatic effects on algorithm performance in the brain age prediction task. For example, the resnet NN algorithm, which had poor $R^2$ in the augmentation-free condition, reaches parity with the baseline model at relatively high augmentation levels ($R^2 = 0.62 \pm 0.016$ standard error of the mean (SEM), red curves). The lstmfcn NN, which also performs poorly with no augmentation, reached even higher $R^2$ than the baseline model with high levels of augmentation ($R^2 = 0.64 \pm 0.009$ SEM, pink curves). However, at these higher levels of augmentation, the data requirements of these two models also increases. Algorithms that were similar in their performance to the baseline in the absence of augmentation improve slightly with the introduction of small amounts of augmentation. For example, the highest $R^2$ reached by any model in these experiments is reached by the blstm1 model at a low value of

augmentation ($R^2 = 0.66 \pm 0.01$ SEM, gray curves). The relatively-simple mlp4 model architecture that does not perform very well in the absence of augmentation, only becomes worse with the introduction of augmentation (blue curves). Further quantification of these trends is laid out in S3 Fig, S4 Fig, S5 Fig.
(PNG)

**S3 Fig. Augmentation effects on best performance.** The black dashed line indicates the $R^2$ of the linear baseline model (PCR Lasso). As seen in S2 Fig, some algorithms only decrease in their performance with increased augmentation (e.g., mlp4, blue curve), but many of the NN algorithms improve their performance with increased augmentation, with some (e.g., resnet, red curve and lstmfcn, pink curve) reaching parity of $R^2$ with PCR Lasso at higher levels of augmentation. At higher levels of augmentation the noise added to the measurements overwhelms all of the useful signal for training and all algorithms perform poorly.
(PNG)

**S4 Fig. Augmentation effects on least accurate performance.** The resilience of the algorithms is quantified for cases where only limited data is available. Almost none of the algorithms, across all augmentation levels, are as resilient to smaller training data as PCR Lasso (dashed line).
(PNG)

**S5 Fig. Rate of performance improvement with increased training sample size.** Here, smaller values indicate more favorable performance (i.e., the algorithm requires a smaller sample size to reach ~63% of its best performance). Several of the algorithms show improved resilience to small sample sizes, relative to the linear baseline (dashed line, PCR Lasso), even under conditions where performance is better than PCR Lasso (e.g., compare blstm1 curve with S1 Fig)
(PNG)

## Author contributions

**Conceptualization:** John Kruper, Adam Richie-Halford, Mareike Grotheer, Jason D. Yeatman, Ariel Rokem.

**Data curation:** John Kruper, Adam Richie-Halford, Mareike Grotheer, Jason D. Yeatman, Ariel Rokem.

**Formal analysis:** John Kruper, Adam Richie-Halford, Joanna Qiao, Asa Gilmore, Kelly Chang, Ethan Roy, Sendy Caffarra.

**Funding acquisition:** Eleftherios Garyfallidis, Serge Koudoro, Theodore D. Satterthwaite, Jason D. Yeatman, Ariel Rokem.

**Investigation:** John Kruper, Adam Richie-Halford, Joanna Qiao, Asa Gilmore, Kelly Chang, Ariel Rokem.

**Methodology:** John Kruper, Adam Richie-Halford, Joanna Qiao, Asa Gilmore, Kelly Chang, Mareike Grotheer, Ethan Roy, Sendy Caffarra, Teresa Gomez, Sam Chou, Matthew Cieslak, Serge Koudoro, Eleftherios Garyfallidis, Theodore D. Sattherthwaite, Jason D. Yeatman, Ariel Rokem..

**Project administration:** Eleftherios Garyfallidis, Theodore D. Satterthwaite, Jason D. Yeatman, Ariel Rokem.

**Resources:** John Kruper, Adam Richie-Halford, Joanna Qiao, Asa Gilmore, Kelly Chang, Mareike Grotheer, Ethan Roy, Sendy Caffarra, Sam Chou, Matthew Cieslak, Serge Koudoro, Eleftherios Garyfallidis, Theodore D. Sattherthwaite, Jason D. Yeatman, Ariel Rokem.

**Software:** John Kruper, Adam Richie-Halford, Joanna Qiao, Asa Gilmore, Kelly Chang, Mareike Grotheer, Ethan Roy, Sendy Caffarra, Teresa Gomez, Sam Chou, Matthew Cieslak, Serge Koudoro, Eleftherios Garyfallidis, Theodore D. Sattherthwaite, Jason D. Yeatman, Ariel Rokem.

**Supervision:** Eleftherios Garyfallidis, Theodore D. Sattherthwaite, Jason D. Yeatman, Ariel Rokem.

**Validation:** John Kruper, Adam Richie-Halford, Joanna Qiao, Asa Gilmore, Kelly Chang.

**Visualization:** John Kruper, Adam Richie-Halford, Joanna Qiao, Asa Gilmore, Kelly Chang.

**Writing – original draft:** John Kruper, Ariel Rokem.

**Writing – review & editing:** John Kruper, Adam Richie-Halford, Joanna Qiao, Asa Gilmore, Kelly Chang, Mareike Grotheer, Ethan Roy, Sendy Caffarra, Teresa Gomez, Sam Chou, Matthew Cieslak, Serge Koudoro, Eleftherios Garyfallidis, Theodore D. Sattherthwaite, Jason D. Yeatman, Ariel Rokem.

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
