## [Decision Letter · Decision Letter 0]

16 May 2025

PCOMPBIOL-D-25-00589

A software ecosystem for brain tractometry processing, analysis, and insight

PLOS Computational Biology

Dear Dr. Rokem,

Thank you for submitting your manuscript to PLOS Computational Biology. After careful consideration, we feel that it has merit but does not fully meet PLOS Computational Biology's publication criteria as it currently stands. Therefore, we invite you to submit a revised version of the manuscript that addresses the points raised during the review process.

Please submit your revised manuscript within 60 days Jul 16 2025 11:59PM. If you will need more time than this to complete your revisions, please reply to this message or contact the journal office at ploscompbiol@plos.org. Please include the following items when submitting your revised manuscript:

We look forward to receiving your revised manuscript.

Kind regards,

Amy Kuceyeski

Academic Editor

PLOS Computational Biology

Hugues Berry

Section Editor

PLOS Computational Biology

**Additional Editor Comments:**

This manuscript has been reviewed by two external experts, and, while mostly positive, there are some clarifications and comments to be addressed before suitable for acceptance. First, please ensure that the code base is usable and has adequate documentation, and second, please make sure to clarify what is new in this manuscript - even if it is to provide comprehensive description of software. Thank you!

**Journal Requirements:**

2) Your manuscript is missing the following sections: Design and Implementation, and Availability and Future Directions. Please ensure that your article adheres to the standard Software article layout and order of Abstract, Introduction, Design and Implementation, Results, and Availability and Future Directions. For details on what each section should contain, see our Software article guidelines:

https://journals.plos.org/ploscompbiol/s/submission-guidelines#loc-software-submissions

Potential Copyright Issues:

- Figure 3. Please confirm whether you drew the images / clip-art within the figure panels by hand. If you did not draw the images, please provide (a) a link to the source of the images or icons and their license / terms of use; or (b) written permission from the copyright holder to publish the images or icons under our CC BY 4.0 license. Alternatively, you may replace the images with open source alternatives. See these open source resources you may use to replace images / clip-art:

5) Please ensure that the funders and grant numbers match between the Financial Disclosure field and the Funding Information tab in your submission form. Note that the funders must be provided in the same order in both places as well.

**Reviewers' comments:**

Reviewer's Responses to Questions

**Comments to the Authors:**

Reviewer #1: This manuscript presents a comprehensive software ecosystem for tractometry analysis based on diffusion MRI, centered on the widely recognized pyAFQ platform. The authors—well-established figures in the field of diffusion imaging and neuroimaging software development—have consolidated multiple efforts into a cohesive, extensible, and open-source pipeline. The manuscript is clearly written, the scope aligns well with the goals of a Software article, and the work has the potential for significant impact given the increasing demand for reproducible and scalable tractometry pipelines.

Strengths:

pyAFQ and its ecosystem are already well adopted by the diffusion MRI community.

The software is open-source, well-maintained, and integrates modern standards (e.g., BIDS, QSIPrep), with a strong emphasis on reproducibility.

The benchmarking and performance evaluations are thorough and demonstrate meaningful improvements.

Minor Suggestions for Revision (optional):

To be honest, this manuscript is already suitable for acceptance. The following are entirely optional suggestions that may help broaden the accessibility and impact of the work, especially for potential users and readers unfamiliar with pyAFQ. I fully understand if the authors choose not to incorporate these suggestions.

Clarify the Machine Learning Section:

The manuscript includes a detailed description of machine learning models (e.g., MLP, CNN, RNN), but the rationale behind model selection and optimization could be further explained. For instance, why did certain architectures (e.g., BLSTM) outperform others? A brief explanation—even qualitative—would help readers appreciate the modeling choices, especially those less familiar with sequence modeling. A small demonstration or case example would also make this section more engaging and informative.

Expand on Use Case Examples:

While age prediction is a strong example, the manuscript could benefit from highlighting additional applications—particularly those in clinical or longitudinal research. Given the availability of disease-related datasets on platforms like OpenNeuro, demonstrating how pyAFQ could be applied to neurological disorders (e.g., ALS, SC2, Stroke...etc.) may further enhance its relevance and utility.

Clarify Statistical vs. Machine Learning Approaches:

The transition between statistical analysis and machine learning modeling felt a bit abrupt. Including a brief summary or schematic contrasting the two (e.g., interpretability vs. predictive power) could help orient readers to the different goals and outcomes of each approach.

Minor Clarifications:

Please clarify whether all benchmarking datasets (e.g., HBN, ALS) were processed exclusively through QSIPrep, or if other preprocessing pipelines were also considered.

Consider including a concise table summarizing the tools in the ecosystem (e.g., pyAFQ, Tractobot, AFQ-Insight) along with a brief description of their respective roles and how they interoperate.

Conclusion:

This is a well-executed and timely software paper that I believe will be of high value to the neuroimaging community. The above suggestions are offered only as potential ways to further increase its clarity and user-friendliness.

Reviewer: FC Yeh

Reviewer #2: The authors present a suite of software tools for generating and analyzing tractometric profiles of major fiber bundles from diffusion MRI data. These tools can take preprocessed diffusion data, perform tractography, assign tracts to known anatomical bundles, compute summary profiles for each bundle based on diffusion properties or other characteristics, visualize these bundles and their quantitative profiles, and apply statistical and deep-learning models to these profiles to quantify group differences or predict subject phenotypes. Raw diffusion data can be handled through an integrated workflow within QSIprep, and the analytical tools can be applied to tractograms generated by other applications as well. The authors provide many example scripts to demonstrate how each step in the processing and analysis can be completed via python and R.

General comments: The set of tools presented are all impressive and certainly very useful for the neuroscientific community. The authors do present a compelling set of results to show that these tools are robust and neuroscientifically meaningful. However, much of this ecosystem has already been presented in the authors’ previous publications, and much of this manuscript appears to be demonstrating aspects of the suite that have been previously demonstrated. It is not clear what portions of this manuscript describe new developments or additions to that work. Were the original tools not interoperable, and now are? Some new additions appear to be the neural network models and certain parallelization and acceleration options. The authors should be more explicit about what specifically is new here, and what is consolidated from existing work.

Much of the emphasis appears to be on the improving the usability and interoperability of these tools. The authors should better guide readers and users through the examples on their “tractometry-ecosystem” github repository. Linking to this repository from the main tractometry.org website would be helpful. Upon arriving at the github README, it would be very helpful to be presented with a description of the available demos/notebooks, with direct links. Furthermore, presenting pre-baked outputs for these notebooks would help the user to better understand the “experience” of using the suite without the very lengthy startup/installation process. I was unable to successfully run the notebooks via either the “binder” or “codeocean” interfaces. The “binder” notebooks all encountered errors due to invalid file paths, and the “codeocean” site had no clear means or instructions to execute the actual notebooks.

Clarification:

In line 275, can the authors clarify what they mean by ‘We encoded the tract node as the “length dimension” in these one-dimensional networks’? In addition to this wording, I also think the related Fig S1 would be easier to understand if (e) showed several metrics or tracts, colored to match the column blocks in f-h, or otherwise maintained some more of the details from the similar Fig 1 in their 2021 paper.

Minor:

Missing reference in line 65.

**Have the authors made all data and (if applicable) computational code underlying the findings in their manuscript fully available?**

Reviewer #1: None

Reviewer #2: Yes

PLOS authors have the option to publish the peer review history of their article (what does this mean?). If published, this will include your full peer review and any attached files.

Reviewer #1: **Yes: **Fang-Cheng Yeh

Reviewer #2: No

**Figure resubmission:**
---

## [Decision Letter · Decision Letter 1]

11 Jul 2025

Dear Dr. Rokem,

We are pleased to inform you that your manuscript 'A software ecosystem for brain tractometry processing, analysis, and insight' has been provisionally accepted for publication in PLOS Computational Biology.

Best regards,

Amy Kuceyeski

Academic Editor

PLOS Computational Biology

Hugues Berry

Section Editor

PLOS Computational Biology

The authors have done a great job responding to the reviewer comments.

Reviewer's Responses to Questions

**Comments to the Authors:**

Reviewer #1: The authors addressed the comment professionally.

Reviewer #2: The authors have addressed my questions and concerns. The presented software ecosystem is a very useful contribution to the field. They have clarified the relevance of this manuscript in the context of existing papers from their group. Finally, the revised website, documentation, and examples provided are intuitive and usable.

**Have the authors made all data and (if applicable) computational code underlying the findings in their manuscript fully available?**

Reviewer #1: Yes

Reviewer #2: Yes

PLOS authors have the option to publish the peer review history of their article (what does this mean?). If published, this will include your full peer review and any attached files.

Reviewer #1: **Yes: **Fang-Cheng Yeh

Reviewer #2: No

---

## [Editor Report · Acceptance letter]

PCOMPBIOL-D-25-00589R1

A software ecosystem for brain tractometry processing, analysis, and insight

Dear Dr Rokem,

I am pleased to inform you that your manuscript has been formally accepted for publication in PLOS Computational Biology. Your manuscript is now with our production department and you will be notified of the publication date in due course.

With kind regards,

Anita Estes
